# Evaluation of patients' satisfaction with bronchoscopy procedure

**Aleksandra Karewicz[1], Katarzyna Faber** [2]*, **Katarzyna Karon[1], Katarzyna Januszewska[1], Joanna Ryl[1], Piotr Korczynski[2], Katarzyna Gorska[2], Marta Dabrowska[2], Rafal Krenke[2]**

**1** Students' Research Group"Alveolus", Medical University of Warsaw, Warsaw, Poland, **2** Department of Internal Medicine, Pulmonary Diseases and Allergy, Medical University of Warsaw, Warsaw, Poland

* kfaber@wum.edu.pl, kfa6er@gmail.com

## Abstract

### Background

The bronchoscopy (BS) experience provokes anxiety amongst some patients. It can have a negative impact on the course of the procedure and on the willingness of patients to undergo the next BS in the future.

### Objective

We aimed to identify factors influencing patients' satisfaction with BS.

### Methods

The prospective study had been conducted between January and June 2019. It included patients hospitalized in our Department, who underwent elective BS. Patients assessed their anxiety and satisfaction level before and after BS using the Visual Analogue Scale (VAS). Data concerning the course of the bronchoscopy was collected.

### Results

The median level of anxiety prior to the procedure was moderate, higher in women ($p<0.0001$). The majority of patients (116/125, 93%) were satisfied with appropriate information before the procedure. Almost one-third of the interviewees (39/125, 31%) declared complete satisfaction (VAS = 0) with their procedure, 17 patients (14%) were dissatisfied (VAS >5/10). Overall 113 (90%) patients declared unconditional consent for future bronchoscopy. Multivariate linear regression analysis revealed two factors affecting patients' satisfaction with bronchoscopy: anxiety prior to BS (standardized regression coefficient $\beta = 0.264$, $p = 0.003$) and discomfort ($\beta = 0.205$, $p = 0.018$). Neither age, degree of amnesia, duration of the procedure nor its type added any significant value as factors affecting patient satisfaction. The most common factors inducing patients' discomfort during BS were local anesthesia of the throat (56/125, 45%) and cough (47/125, 38%).

**Data Availability Statement:** The shared database in is an anonymized version - it doesn't have patients' names, surnames, and PESEL Numbers [Polish acronym for "Universal Electronic System

for Registration of the Population"]. All other data is available in the shared database. For a full database please contact the Science Department of Medical University of Warsaw: the person responsible for our department is Ewa Hieronimczuk, email: ewa.hieronimczuk@wum.edu.pl; phone number: (+48 22) 57 20 190.

**Funding:** The authors received no specific funding for this work.

**Competing interests:** The authors have declared that no competing interests exist.

## Conclusions

Low anxiety level before bronchoscopy and reduced discomfort during the procedure are associated with better patient satisfaction. Thus, it is important to reduce patient anxiety and discomfort during the procedure.

## Introduction

Bronchoscopy (BS) is a common medical procedure used to diagnose or treat patients with a variety of respiratory diseases. Although it is considered safe and severe complications are rare [1], BS may be associated with significant distress and anxiety among patients [2–4]. Anxiety causes increased cortisol levels, blood pressure, heart rate and respiration rate [5, 6], which could influence the course of the procedure and cause an increased number of complications. In addition, discomfort produced by bronchoscopy is mostly related to cough, dyspnea, chest pain or nausea [3, 7]. The level of anxiety experienced by the patients before and during bronchoscopy is determined by numerous factors, including age, gender [8], insufficient information about the aim of the procedure, its course and possible complications [9].

Premedication plays a crucial role in reducing stress-related to any invasive procedures. It is an important issue as stress caused by the fear of bronchoscopy may negatively affect patients' compliance during the procedure and their willingness to undergo potential re-examination in the future [10]. Due to the immense progress in interventional bronchoscopy used for advanced diagnostic and therapeutic purposes, the importance of premedication or anesthesia-related to BS is also growing [11].

Patients' satisfaction with BS is a subjective feeling that depends on patients' expectations and may be influenced by various factors. The level of patient satisfaction is increasingly emphasized as a significant outcome measure for bronchoscopy, along with its diagnostic and therapeutic efficacy. Hence, the knowledge of the factors which affect patients' satisfaction is important in terms of an optimal preparation and conducting the procedure. Therefore, the aim of this study was to identify factors that influence patients' satisfaction with bronchoscopy.

## Material and methods

### General study design

This was a prospective, single-center, observational, cross-sectional study performed in the Department of Internal Medicine, Pulmonary Diseases and Allergy of the Medical University of Warsaw, Poland between January and June 2019. Patients scheduled for an elective bronchoscopy understood as a non-emergency bronchoscopy procedure (diagnostic and therapeutic), scheduled in advance, were included. A written informed consent was obtained from all enrolled patients. The study was approved by the Ethics Committee of the Medical University of Warsaw (AKB/ 234/2018).

### Patients

Patients with different malignant and nonmalignant pulmonary diseases who were admitted for elective diagnostic bronchoscopy were informed about the aim of the study and asked for their consent to participate. Patients were reassured that their disagreement would not change the course of the procedure. The inclusion criteria were: 1) age above 18 years, 2) indication

for an elective bronchoscopy (scheduled non-emergency bronchoscopy procedures) performed on a hospital basis, 3) written informed and voluntary consent to participate in the study. The exclusion criteria were as follows: 1) age under 18 years, 2) urgent interventional bronchoscopy, 3) lack of agreement to participate in the study 4) inability to read, understand, complete surveys, or collaborate with the medical staff; 5) bronchoscopy under general anesthesia, 6) bronchoscopy on the out-patient basis.

## Methods

All patients included in the study were asked to complete two original questionnaires:

1. Questionnaire B (= before) completed the day before the scheduled bronchoscopy.

2. Questionnaire P (= post) completed 24 hours after the procedure.

Questionnaire B included basic demographic data, data on the adequacy of information about the procedure provided by the attending physician and the nursing staff, patient's expectations and the level of fear related to bronchoscopy (see S1 Table). Questionnaire P included questions aimed at assessing patient satisfaction with bronchoscopy (see S2 Table). Both questionnaires were originally created for the purpose of the current project and based on the British Thoracic Society (BTS) guideline for diagnostic flexible bronchoscopy in adults [12]. Satisfaction with bronchoscopy was defined as an overall subjective assessment of impressions and experiences, including complaints related to the procedure. It was measured using a 10 cm Visual Analogue Scale (VAS). Complete satisfaction was rated as 0, while complete dissatisfaction as 10 cm. Both questionnaires included patient identification data (ID) in order to link this data with factors possibly affecting the satisfaction from the procedure.

## Outcome points

**Primary outcome.**   The correlation between patient satisfaction with bronchoscopy and anxiety before the procedure both measured by VAS

**Secondary outcome.**   Identification of factors affecting patient satisfaction with BS

## Bronchoscopy

All patients scheduled for bronchoscopy received routine information on the procedure, including its aim, course, and possible complications, from the attending physician and the nursing staff. Importantly, in order to avoid changes in standards of informing the patients about the procedure, the staff was not informed, which patients were included in the study. Patients who had bronchoscopy performed under general anesthesia were excluded from analysis as general anesthesia may influence patients' satisfaction related to the procedure. Thus, patients who had rigid bronchoscopy were not included in this study. All other types of fiberoptic bronchoscopic procedures were accepted in the study protocol. The oral route was used to introduce all types of bronchoscopes to the lower airways, which is the standard way of bronchoscope insertion in our Department.

The procedures performed during bronchoscopy were as follows: endobronchial secretion removal, bronchoalveolar lavage (BAL), endobronchial forceps biopsy, endobronchial brush biopsy, transbronchial lung biopsy, endobronchial ultrasound and endobronchial ultrasound-guided transbronchial needle aspiration (EBUS-TBNA), endobronchial ultrasound with radial probe and transbronchial biopsy (rEBUS-TBB), and foreign body removal. All bronchoscopies were performed under local anesthesia and conscious sedation (midazolam orally or midazolam and fentanyl intravenously). Sprayed lidocaine (2% and 10%) was applied as a local

anesthetic to the throat. Additional doses of 2% lidocaine were applied to the trachea and bronchi via the working channel of the bronchoscope.

In all patients, bronchoscopy was performed by a pulmonologist with 10–15 years of experience in performing bronchoscopy. In the vast majority of procedures, one or more of the following bronchoscopes was used: video bronchoscope BS-1TH190, EBUS scope BS-UC180F (Olympus, Tokyo, Japan). Patients were monitored closely during the procedure, as well as after the bronchoscopy with the use of a dedicated report form (see S3 Table). All adverse events up to 24 hours after the completion of the procedure were noted by the attending physician in a separate report form (see S4 Table). Criteria of serious adverse events related to BS were adopted from the BTS guideline for diagnostic flexible bronchoscopy in adults [12].

### Statistical analysis and sample size calculation

Power analysis and sample size calculations for correlation analysis showed that a sample size of 85 patients would provide 80% statistical power to detect weak (r = 0.3) correlation (alpha = 0.05, beta = 0.20) [13]. The number of enrolled patients was increased by 15 to allow for a 15% drop-out. Thus, a total number of 100 patients was a minimum required to conduct this study.

As data did not have a normal distribution, non-parametric tests were used. Data are presented as the median and interquartile range (IQR) unless otherwise specified. Differences between satisfied (VAS rating < 1 cm) and unsatisfied patients (VAS rating >5 cm), between different types of bronchoscopy and different types of anesthesia were compared using a chi-square test for categorical variables and Mann Whitney U test for continuous variables. Spearman coefficient was used for correlation analysis. The factors affecting patients' satisfaction with BS were evaluated using correlation analysis, univariate and multivariate linear regression analysis. All parameters (expressed in interval scale) were screened in univariate analysis and selected to build a multivariate linear regression model with backward stepwise analysis. The optimal model was chosen based on the highest adjusted R square value. All analyses were performed using Statistica 13.0 (StatSoft Inc., Tulsa, OK, USA) and MedCalc 13.2.2 (MedCalc Software bvba, Ostend, Belgium). A p-value lower than 0.05 was regarded as significant.

### Results

From 200 patients admitted to our Department for elective bronchoscopy, 157 met the inclusion criteria and agreed to participate in the study. All questionnaires were completed by 125 patients and those patients were included in the final analysis (Fig 1).

There were 67 male (54%) and 58 female patients. The median age was 66 years (IQR 58–73); 32 (26%) patients were never smokers, while 39 (31%) and 54 (43%) were active smokers and ex-smokers, respectively. The median smoking history was 29 pack-years (IQR 10–40). Indications for bronchoscopy and comorbidities are stated in S1 and S2 Figs.

The median level of anxiety prior to bronchoscopy was moderate, i.e. 5/10 cm (IQR 2.5–6) according to VAS. Women were significantly more anxious before the procedure than men [5 (IQR 4.6–8) cm vs 3.5 (IQR 0.8–5) cm, respectively, p<0.0001]. As many as 116 patients (93%) were satisfied with the level of information received before bronchoscopy.

Midazolam was given orally as premedication in 125 patients (100%) (in 123–7.5 mg, in 2 patients -3.25 mg) and in 39 patients (31%) intravenous fentanyl was added (median dose 50 mg, IQR 50–87.5).

Seventy-four patients (59%) had flexible (fiberoptic) bronchoscopy (including 3 procedures via tracheostomy), while EBUS had 51 subjects (41%)–linear, radial or both (linear and radial) EBUS was performed 33, 10 and 8 patients, respectively (Table 1).

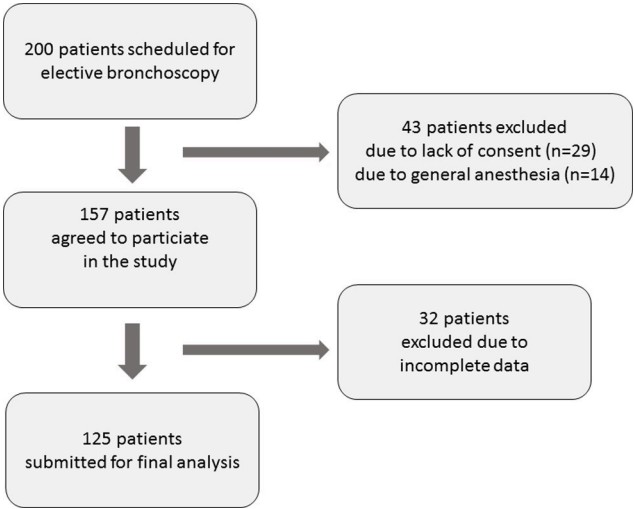

**Fig 1. Participants flow diagram.**

Subjective satisfaction with bronchoscopy was estimated as good by most patients (Fig 2). The median VAS satisfaction rating was 1 cm (IQR 0–4). Thirty-nine patients (31%) declared absolute satisfaction with their procedure (VAS 0/10), while 17 patients (14%) were unsatisfied (VAS > 5/10). The vast majority of patients (n = 113, 90%) declared unconditional consent for future bronchoscopy. There was a positive correlation between satisfaction and willingness for future bronchoscopy (rho = 0.404, p<0.0001). We did not find a difference in satisfaction after BS between younger and older patients (≤ 65 and >65 years of age, respectively) or those who had the first or a repeated BS. There were no differences in either anxiety level before the procedure or satisfaction with it between patients who had video bronchoscopy and those who had EBUS (Table 1). Similarly, no differences were found between patients who were sedated with midazolam alone or both with midazolam and fentanyl (Table 2).

The primary outcome of the study was a weak positive correlation between dissatisfaction with bronchoscopy and the level of anxiety before the procedure (Spearman coefficient rho = 0.276, p = 0.0014) or patients' discomfort during the procedure (rho = 0.309,

**Table 1. Differences between different types of bronchoscopy procedures.**

|  | VBS n = 74 | EBUS n = 51 | P value |
|---|---|---|---|
| Age (years) | 65.5 (60–74) | 66 (55–72) | 0.379 |
| Anxiety prior to BS | 5 (1–5.9) | 5 (3–6) | 0.440 |
| Duration of BS (min) | 9 (6–12) | 20 (15–27) | <0.0001 |
| Any complication after BS (number of patients) | 11 (15%) | 7 (14%) | 0.858 |
| Number of ailments during BS | 1 (1–2) | 2 (1–2) | 0.474 |
| Satisfaction with BS in VAS (cm) | 1 (0–3) | 1 (0–4.3) | 0.658 |
| Unconditional consent for future BS (number of patients) | 70 (95%) | 43 (84%) | 0.055 |

Data is presented as rank/ median and IQR (in parenthesis) or the number of patients and percentage (in parenthesis). Both groups were compared using the chi-squared test for categorical variables and the Mann-Whitney U test by rank for continuous variables.

n = number of patients, VBS—videobronchocopy, EBUS—endobronchial ultrasound, BS–bronchoscopy, VAS–Visual Analogue Scale, F—Female, M-Male

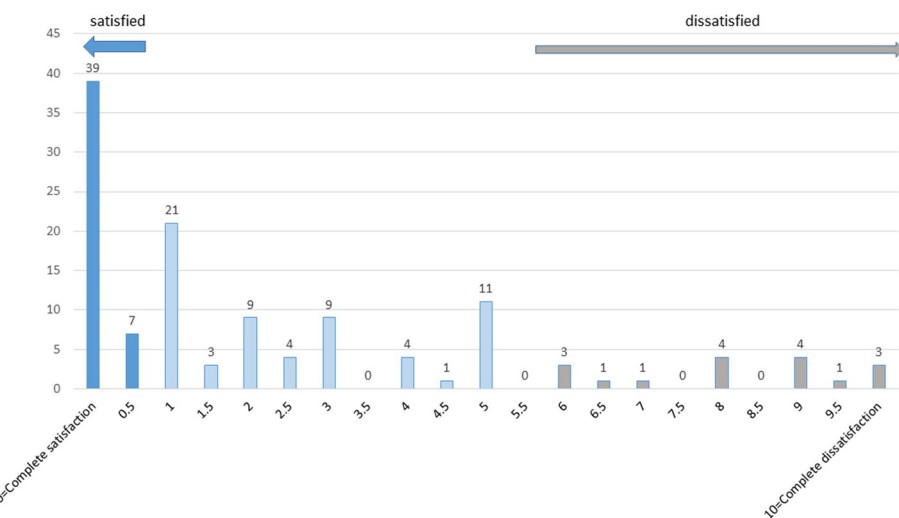

**Fig 2. Distribution of patients' satisfaction with bronchoscopy.**

p = 0.0005). The most common factors inducing patient discomfort during BS were local anesthesia of the upper airways (56/125, 45%) and cough (47/125, 38%), the remaining factors are shown in Fig 3. No other significant correlations between dissatisfaction with BS and other factors were found.

Univariate regression analysis revealed that anxiety prior to bronchoscopy and discomfort during the procedure were the only significant factors affecting patient satisfaction with bronchoscopy (Table 3). Then, multivariate linear regression analysis revealed only two factors affecting patients' satisfaction with bronchoscopy: anxiety prior to BS (standardized regression coefficient β = 0.264, standard error 0.086, p = 0.003) and discomfort (β = 0.205, standard error 0.086, p = 0.018). Neither age, satisfaction with information about the bronchoscopy, its duration nor the degree of amnesia added any significant value as factors affecting patient satisfaction.

**Table 2. Differences between different types of anesthesia.**

|  | Midazolam only n = 86 | Midazolam+ fentanyl n = 39 | P value |
|---|---|---|---|
| Age (years) | 67 | 67 | 0.964 |
|  | (59.5–75) | (58–71) |  |
| Anxiety prior to the BS | 5 | 5 | 0.184 |
|  | (1–5.5) | (4–7.5) |  |
| Duration of BS (min) | 10.5 | 19 | < 0.001 |
|  | (7–16.25) | (11.5–26.5) |  |
| Any complications after BS (number of patients) | 11 (13%) | 8 (21%) | 0.256 |
| Number of ailments during BS | 2 | 1 | 0.961 |
|  | (1–2) | (1–2) |  |
| Satisfaction with BS in VAS | 1 | 2 | 0.833 |
|  | (0–3.75) | (1–5.5) |  |
| Unconditional consent for future | 79 | 36 | 0.932 |
| BS (number of patients) | (92%) | (92%) |  |

Data are presented as median and IQR (in parenthesis) or the number of patients and percentage (in parenthesis). Both groups were compared using the chi-squared test for categorical variables and the Mann-Whitney U test by rank for continuous variables.

n = number of patients, BS–bronchoscopy, VAS- Visual Analogue Scale

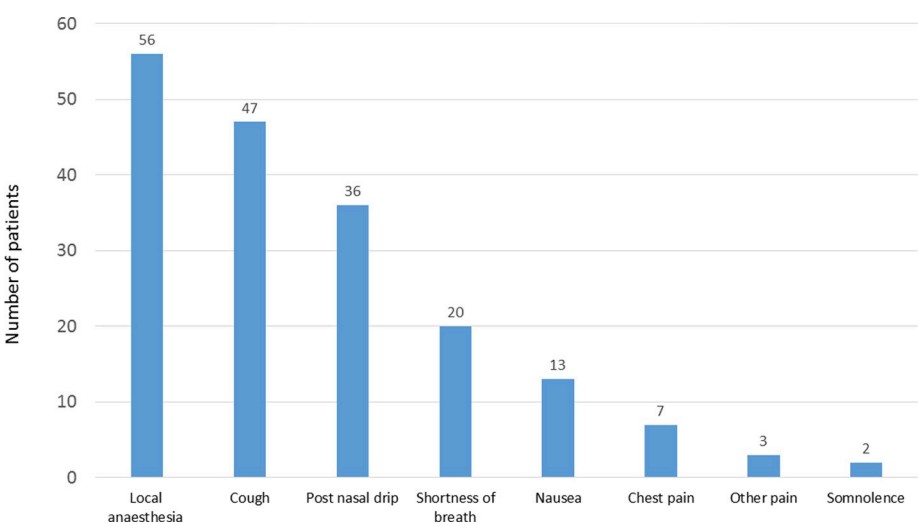

**Fig 3. Factors associated with patients' discomfort during bronchoscopy.**

In 107 patients (86%) no complications during BS were noted. There was one serious adverse event during BS (1/125, 0.8%). The patient presented acute bradypnoea related to the administration of midazolam (<8 breaths/minute) during bronchoscopy and reversal dose of flumazenil was given with immediate positive effect. Other complications during bronchoscopy occurred rarely (17, 14%) and included: moderate bleeding associated with biopsy procedures (8 patients, 6%), fever (5 patients, 4%) transient oxygen desaturation < 85% (2 persons, 3%), COPD exacerbation (1 patient, 0.8%) and transient bradypnoea with spontaneous reversal (1 patient, 0.8%).

Comparison of satisfied and unsatisfied patients showed differences in the level of anxiety prior to bronchoscopy (p = 0.0009), patient discomfort during the procedure (p = 0.046) and not remembering the procedure due to premedication (p = 0.014) (Table 4).

## Discussion

Our study showed that a lower level of anxiety before bronchoscopy and during the procedure is associated with higher patient satisfaction after BS. It emphasizes that an adequate and comprehensive information on the procedure as well as premedication and sufficient anesthesia are important to reduce patient anxiety and discomfort during bronchoscopy. As the level of patient satisfaction is an important outcome measure for BS, the knowledge of the factors

**Table 3. Univariate and multivariate linear regression analysis of the variables associated with patient satisfaction with bronchoscopy.**

| Parameters | Univariate Analyses | | | Multivariate Analysis | | |
|---|---|---|---|---|---|---|
| | Beta | Standard error | *P-value* | Beta | Standard error | *P-value* |
| Anxiety before BS | 0.306 | 0.086 | 0.0006 | 0.264 | 0.086, | 0.003 |
| Discomfort during BF | 0.255 | 0.087 | 0.004 | 0.205 | 0.086 | 0.018 |
| Age of patient | 0.015 | 0.090 | 0.864 | | NI | |
| Duration of BF | 0.039 | 0.092 | 0.674 | | NI | |
| Satisfaction with information about the BF | -0.096 | 0.090 | 0.288 | | NI | |
| Not remembering BF | -0.136 | 0.090 | 0.135 | | NI | |

Abbreviations: NI, not included in the best multivariate linear regression model.

**Table 4. Differences between satisfied and unsatisfied patients.**

| | Satisfied VAS < 1 cm n = 46 | Unsatisfied VAS > 5–10 cm n = 17 | P value |
|---|---|---|---|
| Age (years) | 67 (56–74) | 61 (56–66) | 0.123 |
| Sex (F/M) | 20/26 | 11/6 | 0.135 |
| Smoking history (S/ExS/NS) | 14/22/10 | 6/6/5 | 0.658 |
| Pack-years | 30 (15–40) | 21 (1–34) | 0.369 |
| Being well informed before procedure | 43 (93%) | 17 (100%) | 0.281 |
| Anxiety prior to the bronchoscopy | 3.5 (0–5) | 6 (5–9.5) | 0.0009 |
| Previous bronchoscopy | 12 (26%) | 3 (18%) | 0.485 |
| Type of bronchoscopy | | | |
| VBS | 29 | 10 | 0.793 |
| sEBUS | 11 | 5 | 0.656 |
| rEBUS | 4 | 1 | 0.874 |
| sEBUS+rEBUS | 2 | 1 | 0.680 |
| Duration of bronchoscopy (min) | 14.5 (7–18.5) | 14 (9–25) | 0.535 |
| Midazolam as only premedication | 33 (72%) | 12 (71%) | 0.928 |
| Midazolam+Fentanyl as premedication | 13 (28%) | 5 (29%) | 0.928 |
| Any complication related to BS | 6 (13%) | 6 (35%) | 0.061 |
| Number of factors inducing patient discomfort during BS | 1 (0–2) | 2 (1–3) | 0.046 |
| Not remembering BS | 24 (52%) | 3 (18%) | 0.014 |
| **Procedures during bronchoscopy** | | | |
| Endobronchial secretion removal | 16 (34%) | 7 (41%) | 0.640 |
| Bronchoalveolar lavage | 20 (43%) | 7 (41%) | 0.869 |
| Forceps biopsy | 13 (28%) | 2 (12%) | 0.172 |
| Bronchial brushing | 3 (6%) | 0 | 0.680 |
| EBUS -TBNA | 13 (28%) | 5 (29%) | 0.928 |
| rEBUS -TBB | 4 (9%) | 1 (6%) | 0.714 |

Data are presented as median and IQR (in parenthesis) or the number of patients and percentage (in parenthesis). Both groups were compared using the chi-squared test for categorical variables and the Mann Whitney U test for continuous variables.

n = number of patients, VAS- Visual Analogue Scale, F—female, M-Male, NS—never smoker, S—smoker, Ex—ex-smoker, VBS- videobronchocopy, sEBUS -linear endobronchial ultrasound, rEBUS- radial endobronchial ultrasound, BS -bronchoscopy, EBUS-TBNA- endobronchial ultrasound-guided transbronchial needle aspiration, rEBUS-TBB—radial probe endobronchial ultrasound-guided transbronchial biopsy.

affecting bronchoscopy related patient discomfort is crucial to improve the level of patient tolerance of the procedure.

Our results are consistent with the results of previous studies [14–16]. Lechtzin et al. [15] reported that dissatisfaction after bronchoscopy was associated with anxiety prior to the procedure or its complications. The pre-bronchoscopy level of anxiety was identified as an important factor predicting patient satisfaction during and after the procedure by other authors as well [14, 16]. The results of the study by Yildirim et al. [14] suggested that the operator should adjust the course of BS to the patient's anxiety level, i.e. by choosing the oral route or by shortening the procedure time with fewer interventions if the patient is very anxious. In addition, the operator's experience may also influence the patient's discomfort during BS [15, 16]. Mitsumune et al. [16] suggested that the more the patient is anxious, the more experienced bronchoscopist should perform the bronchoscopy. On the other hand, there have also been studies that found no correlation between the operator's experience and patient satisfaction [17]. This aspect was not addressed in our study, as our aim was to evaluate other factors related to pre-bronchoscopy anxiety level and post-bronchoscopy satisfaction. Thus, all the bronchoscopists participating in the study had a comparable level of expertise.

Similarly to other authors [18], we have found that the female sex was associated with a higher level of anxiety before bronchoscopy. Albeit other authors [10, 14, 17] also reported a lower level of post-bronchoscopy satisfaction in women, this was not the case in our study. Gender differences in anxiety before and dissatisfaction after BS may be associated with gender differences in the perception of pain. Previous studies demonstrated that postoperative or procedural pain may be more severe in females than males [19].

Sun et al. [20] suggested that communication with patients and their education are critical for experiencing satisfaction with BS and readiness to undergo a repeated BS in the future. In our study, 93% of patients were satisfied with the adequate information about upcoming bronchoscopy. Such a high level of satisfaction with the provided information to the patients was certainly related to the elective nature of bronchoscopy applied in an in-patient setting. Some studies showed that the amount of information provided to the patient prior to BS may decrease the level of patient anxiety [2, 10, 15]. However, other studies showed no correlation between the quantity and quality of information given to patients and their satisfaction with the procedure [18, 21] or even demonstrated that patients who were provided with a greater amount of information on possible post-BS complications experienced significantly more anxiety compared with the patients who were less informed [22]. Nonetheless, bronchoscopy guidelines recommend giving patients both verbal and written precise information prior to the procedure [12]. Additionally, it is reasonable to evaluate patient satisfaction after the procedure and address the areas of dissatisfaction and discomfort to improve the quality of the service [12].

Although it is crucial to minimize discomfort during bronchoscopy, it cannot be completely eliminated. In this study, discomfort during bronchoscopy obviously affected patient satisfaction with BS. Furthermore, a correlation between dissatisfaction and discomfort was weak, but higher than with anxiety before the procedure. Among our patients, local anesthesia of the upper airways and cough were the most common factors that caused the discomfort. These results correspond to those reported by Fujimoto et al. [10] who also pointed out throat anesthesia as the most frequent cause of BS-related patient discomfort. Hence, it is crucial to inform patients about this unpleasant but important part of the procedure, which allows the insertion of the bronchoscope to the lower airways, decreases the severity of cough and throat pain. Warning of the patient beforehand is critical, as the unexpected discomfort was shown to enhance patient reluctance to undergo a re-examination [10]. Other factors related to patient discomfort in our study were cough, postnasal drip and shortness of breath. The better the control of cough, pain and shortness of breath, the more eager the patient is to return for a repeated BS [15].

Conscious sedation is regarded as an important factor that may reduce anxiety and improve the level of patient satisfaction with the procedure [23, 24]. In our study, conscious sedation was used in all patients. We found that satisfied patients declared amnesia more frequently than unsatisfied subjects, however, we did not document any significant correlation between the level of amnesia and patients' satisfaction with bronchoscopy. Irrespective of the type of sedation, it is important to monitor the sedation effects and titrate the dose of sedatives to avoid respiratory depression [12].

Another factor that may influence patient satisfaction with bronchoscopy are complications related to the procedure. However, we did not observe such a relationship in our study, which may have resulted from a relatively low percentage of complications, noted in solely 18/125 patients (14%). Only one severe adverse event was noted in our group. Severe complications during bronchoscopy are very rare, and, in some cases cannot be prevented [1, 3]. However, it is vital to minimize the risk of complications, such as bleeding, oxygen desaturation, arrhythmia or bradypnoea. The experience of the operator performing the bronchoscopy is an important factor for lowering the number of complications [25, 26].

Some factors that have been previously reported to affect patient anxiety and satisfaction with bronchoscopy were not confirmed in our study. Those include patient age and previous bronchoscopies. Data on the relationship between patient age and anxiety are ambiguous. Poi et al. [18] demonstrated that younger patients were more "fearful", whereas Aljohaney [8] found that older patients had a significantly higher anxiety score. According to Hehn et al. [27], elderly patients tolerate bronchoscopy as well as younger ones. Andrychiewicz et al. [21] showed that patients who had undergone BS in the past had a significantly better understanding of the rationale for performing the procedure and the type of procedure planned and anxiety was reported significantly more often in patients undergoing BS for the first time.

We are aware of several limitations in our study. Firstly, as it was a single-center study, the results may be specific for our facility and the institutional standards for performing bronchoscopy. The results are certainly highly dependent on the methods of sedation during bronchoscopy, throat anesthesia and preceding information of the examination as well as on the profile of patients including their cultural or ethnic backgrounds. Patients who were awaiting an elective bronchoscopy understood as a non-emergency bronchoscopy procedure (diagnostic and/ or therapeutic), scheduled in advance, were included. Secondly, the study group was rather small due to single-center study, inclusion criteria and short duration of the study. Thirdly, we included only in-patients treated in our Department, which may have been associated with a selection bias and our results may not necessarily be extrapolated to BF performed in out-patients. Furthermore, although the questionnaires used in the study were checked for their comprehensibility, we did not validate them in advance, which is a limitation of this study. Next, as we did not use sedation score during the study it is difficult to estimate the influence of sedation on satisfaction with BS precisely. Finally, the sample size calculation was based on an assumption of linear Pearson correlation, but finally, we used Spearman correlation due to lack of normal distribution of data. Despite all these limitations, we believe that the results of our study are important as they indicate how essential it is to make an effort to reduce patient anxiety before BS and discomfort during it.

## Conclusions

Low patient anxiety before bronchoscopy and reduced discomfort are associated with higher patient satisfaction after the procedure. Adequate and comprehensive information on the aim and course of bronchoscopy, as well as premedication, allow to reduce patient anxiety and discomfort during the procedure and thus increase their post-bronchoscopy satisfaction.

## Supporting information

**S1 Table. Questionnaire B (before bronchoscopy).**
(PDF)

**S2 Table. Questionnaire P (post bronchoscopy).**
(PDF)

**S3 Table. Course of bronchoscopy.**
(PDF)

**S4 Table. Physician report after bronchoscopy.**
(PDF)

**S1 Fig. Indications for bronchoscopy.**
(TIF)

**S2 Fig. Comorbidities in population of the study.**
(TIF)

**S1 Data.**
(XLSX)

## Acknowledgments

The authors would like to thank Marta Maskey-Warzęchowska MD, PhD and Katarzyna Mycroft MD for their editorial assistance and manuscript review.

## Author Contributions

**Conceptualization:** Marta Dabrowska, Rafal Krenke.

**Formal analysis:** Marta Dabrowska.

**Investigation:** Aleksandra Karewicz, Katarzyna Faber, Katarzyna Karon, Katarzyna Januszewska, Joanna Ryl, Piotr Korczynski, Katarzyna Gorska, Marta Dabrowska.

**Supervision:** Marta Dabrowska.

**Visualization:** Katarzyna Faber, Marta Dabrowska.

**Writing – original draft:** Aleksandra Karewicz, Katarzyna Faber, Katarzyna Karon, Katarzyna Januszewska, Joanna Ryl, Marta Dabrowska.

**Writing – review & editing:** Aleksandra Karewicz, Katarzyna Faber, Katarzyna Karon, Katarzyna Januszewska, Joanna Ryl, Piotr Korczynski, Katarzyna Gorska, Marta Dabrowska, Rafal Krenke.

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
