## [Decision Letter · Decision Letter 0]

18 Aug 2021

PONE-D-21-03696

Evaluation of patients’ satisfaction with bronchoscopy procedure.

PLOS ONE

Dear Dr. Faber,

Thank you for submitting your manuscript to PLOS ONE. After careful consideration, we feel that it has merit but does not fully meet PLOS ONE’s publication criteria as it currently stands. Therefore, we invite you to submit a revised version of the manuscript that addresses the points raised during the review process.

The reviewers have identified several aspects of your methodological design and statistical analyses that will require further clarification in order to evaluate fulfilment of the journal's publication criteria. Please respond carefully to each of the points they have raised when preparing your revisions.

We look forward to receiving your revised manuscript.

Kind regards,

Jamie Males

Staff Editor

PLOS ONE

Journal Requirements:

2. Please include additional information regarding the survey or questionnaire used in the study and ensure that you have provided sufficient details that others could replicate the analyses. For instance, if you developed a questionnaire as part of this study and it is not under a copyright more restrictive than CC-BY, please include a copy, in both the original language as well as the English version already provided, as Supporting Information.

"No" 

"No"

6. Please upload a new copy of Figure 2 as the detail is not clear. Please follow the link for more information: https://blogs.plos.org/plos/2019/06/looking-good-tips-for-creating-your-plos-figures-graphics/" https://blogs.plos.org/plos/2019/06/looking-good-tips-for-creating-your-plos-figures-graphics/.

Reviewers' comments:

Reviewer's Responses to Questions

**Comments to the Author**

1. Is the manuscript technically sound, and do the data support the conclusions?

Reviewer #1: Partly

Reviewer #2: Yes

2. Has the statistical analysis been performed appropriately and rigorously? 

Reviewer #1: Yes

Reviewer #2: Yes

3. Have the authors made all data underlying the findings in their manuscript fully available?

Reviewer #1: No

Reviewer #2: Yes

4. Is the manuscript presented in an intelligible fashion and written in standard English?

Reviewer #1: Yes

Reviewer #2: No

5. Review Comments to the Author

Reviewer #1: This study reported patient satisfaction with elective bronchoscopy and factors related to the bronchoscopy-related anxiety and satisfaction after the procedure. Authors found that low patient anxiety before bronchoscopy, less discomfort during the procedure, amnesia due to premedication and less complications are associated with a higher patient satisfaction after bronchoscopy. However, multivariate regression analysis revealed anxiety prior to BS (standardized regression coefficient β=0.243, p=0.003), discomfort (β=0.186, p=0.021) and complications (β=0.239, p=0.003) as the only significant factors affecting patient satisfaction with bronchoscopy.

There are some issues as below.

Major points

1. On primary endpoint

Authors aimed to identify factors influencing patients’ satisfaction with BS. However, they did sample size calculations for correlation analysis to show that a sample size of 85 patients would provide 80% statistical power to detect weak (r=0.3) correlation (alpha = 0.05, beta = 0.20). In general, sample size calculation should be used to clarify the main objective. I wonder why they use correlation analysis to detect weak (r=0.3) correlation. Furthermore, did they do this sample size calculation in order to find out which factors correlated with which factors?

2. Among the results of this manuscript, I think the result of multivariate regression analysis which described in line 246 is most important information. Authors concluded this result in the Conclusions session in line 379. However, they included “amnesia due to premedication” in Discussion session in line 276 to 278. If they included “amnesia due to premedication” from the results of Table 3, I think that “amnesia due to premedication” should be deleted because Table 3 included lots of confounds.

3. I think the result of multivariate regression analysis which described in line 246 should be presented by additional table.

4. In Statistical analysis and sample size calculation, the authors should present more detail on methods of multivariate regression analysis. That is, how they excluded the dependent factors from the selected factors and how they chose the best model among candidate models as final multivariate model.

5. In this study, questionnaire S1and S2 are very important. Questionnaire S1 include only two VAS on anxiety Q7, Q8. Questionnaire S2 include only one VAS on anxiety Q1. Authors used VAS for only anxiety scale. Other items were type of multiple-choice question. In case of multiple-choice question or Likert type question, they should validate them in order to detect the appropriate answers. Therefore, authors should give us the results of validation on these items.

Minor points

1) Line 74; “it is safe and severe complications are rare [1],”→Reference No.1 is too old. Please replace recent reference and recent information.

2) Line 78; “bronchoscopy is most commonly related to related to cough, dyspnea, chest pain or nausea” Is under lined part misprinted or not? Please correct the under lined part.

3) Line 99; Does “elective bronchoscopy” mean “only diagnostic bronchoscopy”? If this is correct it should be defined in the text at first use.

4) Line 108; Authors mentioned “The inclusion to the study was not limited by the type of the procedure”. Considering statistical view, I think they should balance the type of the procedure when they enrolled patients.

5) Line 129; Outcome points had two points. Which is primary endpoint? Which is secondary endpoint?

6) Line 136; “Importantly, the staff had not been informed which patients were included to the study.” Why had the staff not been informed? Please explain this is important.

7) Line 157 and 159; I think Table S3 and S4 are the clinical report form for bronchoscopists and attending physicians not questionnaire.

8) Line 182 to 193; Authors mentioned background characteristics in this part. Please provide the table of “background characteristics” in text or Supporting Information.

9) Line 200 to 204; “Rigid bronchoscopy was performed in 6 patients (4%)”. Therefore, it would better that authors should compare only between flexible (fiberoptic) bronchoscopy without EBUS and that without EBUS.

10) In Table 1 “Satisfaction with BS in VAS”; Please consider the significant digits. “0 – 4.375” means they measured the degree of tens μm.

11) Line 215 to 217 and 337; How many patients achieved the conscious sedation? Depth of anesthesia is very important to evaluation the satisfaction with BS. Please provide depth of sedation by use of sedation score.

12) Line 221; Authors defined VAS > 6/10 as “very unsatisfied”. However, in Table 3 and Figure 2, they defined VAS ≥ 5-10 as “unsatisfied”. I think there is discrepancy. Please explained it. Furthermore, how did they define these cutoff-value?

13) Line 223 and 238 to 240; Was “correlation between satisfaction and willingness for future bronchoscopy” basis for setting for sample size calculations? Or was “correlation between dissatisfaction with bronchoscopy and the level of anxiety before the procedure” basis for setting for sample size calculations? Or was “correlation between dissatisfaction with bronchoscopy and patients discomfort during the procedure” basis for setting for sample size calculations? Which of them were basis for setting for sample size calculations? Was it primary endpoint?

14) Line 371; “Secondly, the study group was rather small due to single-center study, inclusion criteria and short period of study, however the number of included patient was based on sample size calculation.” If authors set the main objective as sample size calculation, sample size of the study group would seem to be more appropriate size. Why did they use correlation analysis to detect weak (r=0.3) correlation?

Reviewer #2: The authors have done a good work in trying to identify the factors that affect patient satisfaction during bronchoscopy in their study titled "Evaluation of patients’ satisfaction with bronchoscopy procedure". The topic and contents have been appropriately dealt with but the authors need to do some minor revisions for the article to be more suitable. First of all, there are many grammatical errors in the text that needs to be corrected. Secondly, the authors in justifying the use of oral route for bronchoscopy, made a statement of the nasal route being causing more discomfort. This justification is not accurate and the authors should just limit their statement to oral route being their preference. Thank you

6. PLOS authors have the option to publish the peer review history of their article (what does this mean?). If published, this will include your full peer review and any attached files.

Reviewer #1: **Yes: **Yuichiro Takeda

Reviewer #2: **Yes: **Adamu Issaka

---

## [Author Response · Author response to Decision Letter 0]

27 Oct 2021

We have applied the changes requested both by the editorial staff and the reviewers. We shared an updated cover letter covering the requested funding and COI information. We also shared a file with detailed responses to all of the reviewers' questions and remarks - see file Response to Reviewers.

---

## [Editor Report · Decision Letter 1]

8 Nov 2021

PONE-D-21-03696R1Evaluation of patients’ satisfaction with bronchoscopy procedure.PLOS ONE

Dear Dr. Katarzyna Faber,

Thank you for submitting your manuscript to PLOS ONE. After careful consideration, we feel that it has merit but does not fully meet PLOS ONE’s publication criteria as it currently stands. Therefore, we invite you to submit a revised version of the manuscript that addresses the points raised during the review process.

We look forward to receiving your revised manuscript.

Kind regards,

Yuichiro Takeda, M.D., Ph.D.

Academic Editor

PLOS ONE

Additional Editor Comments:

There are still some issues as below.

1, Primary outcome: The authors set “a relationship (correlation) between patients’ anxiety and satisfaction with bronchoscopy as the primary outcome. They based their sample size calculation on a study by Bujang MA and Baharum N and provided this article to references. In this provided reference, Bujang MA and Baharum N used Pearson coefficient for correlation analysis. Did they check null hypothesis is equal to zero in this study population? Even if null hypothesis is equal to zero, I wonder why they use Mann Whitney U test or Kruskal- Wallis test that are non-parametric test. Why did they delete Kruskal- Wallis test in Statistical analysis of revised manuscript?

2, Primary outcome: I guess the authors described the result of primary outcome in Line 274 of Manuscript - revised manuscript, clean version.

The authors should describe this study met the primary outcome in Results or Discussion part.

3, Author should describe how to perform the multivariate analysis in this study more detail in in Statistical analysis part. what parameters were they screening by univariate analyses, selection criteria of parameters, how to find independent parameters, how to select best model. Please describe this process in Statistical analysis part.

4. In revised manuscript, you mention only regression model in univariate and multivariate. There are lots of regression method. I think you should not delete “logistic”.

5. In general, univariate analyses are only the screening test for model construction. Table 3 should include multivariate model like below table. However, you should check statistical and clinical independency between “Anxiety before BS” and “Discomfort during what”.

Parameters Univariate Analyses Multivariate Analysis

Beta Standard error P-value Beta Standard error P-value

Anxiety before BS 0.306 0.086 0.0006 0.264 0.086, 0.003

Discomfort during ? 0.255 0.087 0.004 0.366, 0.087 0.00017

Age of patient 0.015 0.090 0.864 NI

Duration of BF 0.039 0.092 0.674 NI

Satisfaction with

information about the BF-0.096 0.090 0.288 NI

Not remembering BF -0.136 0.090 0.135 NI

Abbreviations: NI, not included in the best multivariate logistic regression model.

6. Table 4 is “Differences between satisfied and unsatisfied patients” Why did not authors check all parameters in table 4 by univariate analyses? I think authors should be screening all parameters in table 4. And then they construct some models for the multivariate models. After that, they choose best model and analyze this model as the result. How is it?

7. Although author added the limitation, it is serious issue that they did not validate questionnaires used in the study.

8. There are lots of reports that depth of anesthesia is very important to evaluation the satisfaction with BS. It is serious issue that they did not use sedation score during the study. At least, they should add this point to the study Limitations.
---

## [Author Response · Author response to Decision Letter 1]

23 Dec 2021

A file 'Response to Reviewers' has been sent to 'Attached Files'. Below I've copied the text from that file:

Warsaw, 22th December 2021

Dear Editor and Reviewers,

Thank you for reviewing our manuscript PONE-D-21-03696, titled Evaluation of patients’ satisfaction with bronchoscopy procedure. We appreciate Your comments which have certainly helped us improve the manuscript’s quality. The revised version of the manuscript re-submitted for reevaluation includes changes made strictly according to Your suggestions. We believe, all the points raised by the reviews were addressed. We would be grateful for re-considering our manuscript for publication.

Below, we include the specific responses to the Reviewers’ comments with the hope that they will find them adequate.

• Primary outcome: The authors set “a relationship (correlation) between patients’ anxiety and satisfaction with bronchoscopy as the primary outcome. They based their sample size calculation on a study by Bujang MA and Baharum N and provided this article to references. In this provided reference, Bujang MA and Baharum N used Pearson coefficient for correlation analysis. Did they check null hypothesis is equal to zero in this study population? Even if null hypothesis is equal to zero, I wonder why they use Mann Whitney U test or Kruskal- Wallis test that are non-parametric test. 

Thank you for this comment and suggestion. The manuscript by Bujang MA and Baharum N is a sample size guideline for Pearson correlation coefficient. However, when estimating for sample size, we could not predict whether the assumption for Pearson correlation would be met or not. Thus, our sample size analysis was based on estimation by Bujang et al. 

In our study neither the satisfaction from bronchoscopy (BS) nor anxiety before BS have a normal distribution of values (line 203). Thus we changed the correlation coefficient for Spearman (line 209 and 287, 288) and we used non- parametric tests.

• Why did they delete Kruskal- Wallis test in Statistical analysis of revised manuscript? 

Thank you for this remark. As we had excluded patients who had BS under general anesthesia, we compared only two groups (satisfied vs unsatisfied; midazolam vs midazolam +fentanyl; VBS vs EBUS) using Mann-Whitney U test ( Table 1,2,4). 

• Primary outcome: I guess the authors described the result of primary outcome in Line 274 of Manuscript - revised manuscript, clean version. The authors should describe this study met the primary outcome in Results or Discussion part. 

Thank you for this valuable comment. We changed the text (line 285) to emphasized the primary outcome of the study.

• Author should describe how to perform the multivariate analysis in this study more detail in in Statistical analysis part. what parameters were they screening by univariate analyses, selection criteria of parameters, how to find independent parameters, how to select best model. Please describe this process in Statistical analysis part.

Thank you for this comment. We screened all parameters from the table 4 (that were expressed in interval scale) in univariate analysis. Then we build multivariate linear regression model with backward stepwise analysis with all parameters and choose the model with optimal adjusted R square value (line 215-217).

• In revised manuscript, you mention only regression model in univariate and multivariate. There are lots of regression method. I think you should not delete “logistic”.

Thank you for this issue. We corrected the description of statistical methods for multivariate linear regression analysis (line 215-217).

• In general, univariate analyses are only the screening test for model construction. Table 3 should include multivariate model like below table. 

Parameters Univariate Analyses Multivariate Analysis

 Beta Standard error P-value Beta Standard error P-value

Anxiety before BS 0.306 0.086 0.0006 0.264 0.086 0.003

Discomfort during procedure 0.255 0.087 0.004 0.205 0.086 0.018

Age of patient 0.015 0.090 0.864 NI

Duration of BF 0.039 0.092 0.674 NI

Satisfaction with

information about the BF -0.096 0.090 0.288 NI

Not remembering BF -0.136 0.090 0.135 NI

Abbreviations: NI, not included in the best multivariate logistic regression model. 

Thank you for your comment and suggestion, which is very helpful. We changed Table 3 according to your proposal. 

• However, you should check statistical and clinical independency between “Anxiety before BS” and “Discomfort during procedure”. 

We found very low non-significant correlation between anxiety prior to BS and discomfort during the procedure (r=0.173, p=0.055), so we used both factors in model of multivariate regression analysis. 

• Table 4 is “Differences between satisfied and unsatisfied patients” Why did not authors check all parameters in table 4 by univariate analyses? I think authors should be screening all parameters in table 4. And then they construct some models for the multivariate models. After that, they choose best model and analyze this model as the result. How is it?

Thank you for your suggestion. We screened all parameters with univariate analysis finding only two significant parameters (anxiety before BS and discomfort during procedure), which were selected for multivariate logistic regression analysis. 

• Although author added the limitation, it is serious issue that they did not validate questionnaires used in the study.

Thank you for your comment. Indeed, it is an important issue. We emphasized it in limitation of the study (line 489)

• . There are lots of reports that depth of anesthesia is very important to evaluation the satisfaction with BS. It is serious issue that they did not use sedation score during the study. At least, they should add this point to the study Limitations. 

Thank you for your important remark. We added it to limitations of the study (line 490-491)

---

## [Editor Report · Decision Letter 2]

26 Dec 2021

PONE-D-21-03696R2Evaluation of patients’ satisfaction with bronchoscopy procedure.PLOS ONE

Dear Dr. Faber,

Thank you for submitting your manuscript to PLOS ONE. After careful consideration, we feel that it has merit but does not fully meet PLOS ONE’s publication criteria as it currently stands. Therefore, we invite you to submit a revised version of the manuscript that addresses the points raised during the review process.

We look forward to receiving your revised manuscript.

Kind regards,

Yuichiro Takeda, M.D., Ph.D.

Academic Editor

PLOS ONE

Journal Requirements:

Additional Editor Comments:

I think your manuscript improved now.

Minor points

1) “In our study neither the satisfaction from bronchoscopy (BS) nor anxiety before BS have a normal distribution of values (line 203). Thus we changed the correlation coefficient for Spearman (line 209 and 287, 288) and we used non-parametric tests.”

Does this mean sample size calculation was incorrect in your study? You should include this information as a limitation. This is very important.

2) “The primary outcome of the study was a week positive correlation between dissatisfaction with bronchoscopy and the level of anxiety before the procedure (Spearman coefficient r=0.276, p=0.0014) or patients’ discomfort during the procedure (r=0.309, p=0.0005)” Although the correlation between dissatisfaction with bronchoscopy and the level of anxiety before the procedure was statistically significant, coefficient rho was 0.276 that was below 0.3. On the other hand, the correlation between dissatisfaction with bronchoscopy and patients’ discomfort during the procedure was statistically significant and its coefficient rho was 0.309 that was beyond 0.3. Is the latter the only one that met the main outcome? You should explain this result in the discussion part.

3) I think the coefficient is generally rho (ρ) in the Spearman rank test.

4) Although PLOS one is an open-access journal, the discussion part is slightly long. Can you summarize it?

5) I have a comment for your future study. There is a book that explained how to use Multivariable Analysis: A Practical Guide for Clinicians and Public Health Researchers Second edition by Mitchell H. Katz. He said as below: Whenever possible, "do not use variable selection technique." This is because there is a danger that any variable selection method will select confounders into the model and remove variables that are causally related to the outcome. Also, in both the forward and the backward selection, each variable is evaluated individually, so there is a possibility that two variables that start out as a set and have an important effect on the outcome will not be selected as a set in the final model. There is also the possibility that a variable that is very important in explaining the outcome may not be selected for the model because it is related to a variable that has already been adopted in the model. My opinion is the same as above.

6) This manuscript still has some grammatical errors in the text that needs to be corrected. Please check English in the text carefully.
---

## [Author Response · Author response to Decision Letter 2]

15 Feb 2022

Warsaw, 15th February 2022

Dear Editor and Reviewers,

Thank you for reviewing our manuscript PONE-D-21-03696, titled Evaluation of patients’ satisfaction with bronchoscopy procedure once again. We truly appreciate Your comments which have helped us improve the manuscript’s quality. We believe that all the points raised by the reviewer were addressed. We would be grateful for re-considering our manuscript for publication.

Below, we include the specific responses to the Reviewers’ comments with the hope that they will find them adequate.

1) “In our study neither the satisfaction from bronchoscopy (BS) nor anxiety before BS have a normal distribution of values (line 203). Thus we changed the correlation coefficient for Spearman (line 209 and 287, 288) and we used non-parametric tests.”

Does this mean sample size calculation was incorrect in your study? You should include this information as a limitation. This is very important.

Thank you for this remark. We added this information in limitations of the study (line 201-223). 

2) “The primary outcome of the study was a week positive correlation between dissatisfaction with bronchoscopy and the level of anxiety before the procedure (Spearman coefficient r=0.276, p=0.0014) or patients’ discomfort during the procedure (r=0.309, p=0.0005)” Although the correlation between dissatisfaction with bronchoscopy and the level of anxiety before the procedure was statistically significant, coefficient rho was 0.276 that was below 0.3. On the other hand, the correlation between dissatisfaction with bronchoscopy and patients’ discomfort during the procedure was statistically significant and its coefficient rho was 0.309 that was beyond 0.3. Is the latter the only one that met the main outcome? You should explain this result in the discussion part.

Thank you for your suggestion. We emphasized that correlations were weak, but according to your suggestion we drew more attention to the correlation between dissatisfaction and discomfort (line 406-407). We tried to discuss the significance of discomfort as a factor influencing patients satisfaction with bronchoscopy in detail (line 400-421).

3) I think the coefficient is generally rho (ρ) in the Spearman rank test.

Thank you for this remark. We corrected it for rho (line 286-287).

4) Although PLOS one is an open-access journal, the discussion part is slightly long. Can you summarize it?

Thank you for this hint. We tried to shorten the discussion.

5) I have a comment for your future study. There is a book that explained how to use Multivariable Analysis: A Practical Guide for Clinicians and Public Health Researchers Second edition by Mitchell H. Katz. He said as below: Whenever possible, "do not use variable selection technique." This is because there is a danger that any variable selection method will select confounders into the model and remove variables that are causally related to the outcome. Also, in both the forward and the backward selection, each variable is evaluated individually, so there is a possibility that two variables that start out as a set and have an important effect on the outcome will not be selected as a set in the final model. There is also the possibility that a variable that is very important in explaining the outcome may not be selected for the model because it is related to a variable that has already been adopted in the model. My opinion is the same as above.

Thank you for this remark. We appreciate all comments and suggestion concerning the statistical analysis. We will certainly read this book and use this knowledge in future studies. 

6) This manuscript still has some grammatical errors in the text that needs to be corrected. Please check English in the text carefully.

Thank you for this suggestion. We checked and corrected the manuscript.

---

## [Editor Report · Decision Letter 3]

21 Feb 2022

PONE-D-21-03696R3Evaluation of patients’ satisfaction with bronchoscopy procedure.PLOS ONE

Dear Dr. Katarzyna,

Thank you for submitting your manuscript to PLOS ONE. After careful consideration, we feel that it has merit but does not fully meet PLOS ONE’s publication criteria as it currently stands. Therefore, we invite you to submit a revised version of the manuscript that addresses the points raised during the review process.

We look forward to receiving your revised manuscript.

Kind regards,

Yuichiro Takeda, M.D., Ph.D.

Academic Editor

PLOS ONE

Journal Requirements:

Additional Editor Comments (if provided):

Still, a few issues were raised in your manuscript.

#1; In Manuscript – revised version. February 2022.

Line 260 to 262; “There was a positive correlation between satisfaction and willingness for future bronchoscopy (r=0.487, p<0.0001)”.

I wonder if this “r” is Spearman coefficient rho or not. Please correct or explain it.

#2; And you also need clear and correct Figures and Tables without any misprints (“diabetes; diabetes mellitus,” “arhytmia; arrhythmia,” “asthma; bronchial asthma” in Supplementary Figure 2 and so on) or any additional linear (Q1 in S2 table and so on) to publish. Therefore, you should check all figures and tables and supplementary tables and figures again.

#3; According to Submission Guidelines, manuscripts must be submitted in English. Your supporting information included the Polish version of the S1 Table to S4 Table. Therefore, you should delete them.
---

## [Author Response · Author response to Decision Letter 3]

21 Mar 2022

Dear Editor and Reviewers,

Thank you for reviewing our manuscript PONE-D-21-03696, titled Evaluation of patients’ satisfaction with bronchoscopy procedure once again. We are grateful for Your comments and suggestions which have helped us improve the manuscript’s quality. We hope that all the points raised by the reviewer were addressed. We would be grateful for re-considering our manuscript for publication.

Below, we included the specific responses to the Reviewers’ comments with the hope that they will find them adequate.

#1; In Manuscript – revised version. February 2022.

Line 260 to 262; “There was a positive correlation between satisfaction and willingness for future bronchoscopy (r=0.487, p<0.0001)”.

I wonder if this “r” is Spearman coefficient rho or not. Please correct or explain it.

Response:

Thank you for this apt remark. In line 260-262 Pearson correlation r=0.487, p< 0.0001 was given, but as distribution of data were not normal, we changed it for Spearman correlation rho= 0.404, p<0.0001 (p=0.000003).

#2; And you also need clear and correct Figures and Tables without any misprints (“diabetes; diabetes mellitus,” “arhytmia; arrhythmia,” “asthma; bronchial asthma” in Supplementary Figure 2 and so on) or any additional linear (Q1 in S2 table and so on) to publish. Therefore, you should check all figures and tables and supplementary tables and figures again.

Response:

Thank you for a thorough review. We have revised the supplementary files and tabels and provided the required changes.

#3; According to Submission Guidelines, manuscripts must be submitted in English. Your supporting information included the Polish version of the S1 Table to S4 Table. Therefore, you should delete them.

Respone:

Thank you for this remark. Files in Polish have been deleted.

Thank you for those suggestions. 

 Yours sincerely

 Katarzyna Faber

 Marta Dąbrowska

---

## [Decision Letter · Decision Letter 4]

28 Aug 2022

Evaluation of patients’ satisfaction with bronchoscopy procedure.

PONE-D-21-03696R4

Dear Dr. Faber,

We’re pleased to inform you that your manuscript has been judged scientifically suitable for publication and will be formally accepted for publication once it meets all outstanding technical requirements.

Kind regards,

Yuichiro Takeda, M.D., Ph.D.

Guest Editor

PLOS ONE

Additional Editor Comments (optional):

Through several revisions, I think your manuscript has improved.

ACADEMIC EDITOR:

congratulations to the authors and thanks to the reviewers for the suggestions provided which really helped improve the quality of the manuscript

Silvia Fiorelli

Plos One

Academic Editor

---

## [Editor Report · Acceptance letter]

19 Sep 2022

PONE-D-21-03696R4 

Evaluation of patients’ satisfaction with bronchoscopy procedure. 

Dear Dr. Faber:

I'm pleased to inform you that your manuscript has been deemed suitable for publication in PLOS ONE. Congratulations! Your manuscript is now with our production department. 

Kind regards, 

on behalf of

Dr. Silvia Fiorelli 

Academic Editor

PLOS ONE